# Gastric Neuroendocrine Tumors (g-NETs): A Systematic Review of the Management and Outcomes of Type 3 g-NETs

**DOI:** 10.3390/cancers15082202

**Published:** 2023-04-08

**Authors:** Alice Laffi, Andrea Gerardo Antonio Lania, Alberto Ragni, Valentina Di Vito, Alessia Liccardi, Manila Rubino, Franz Sesti, Annamaria Colao, Antongiulio Faggiano

**Affiliations:** 1Hematology and Oncology, IRCCS Humanitas Research Hospital, Via Manzoni 56, Rozzano, 20089 Milan, Italy; 2Department of Biomedical Sciences, Humanitas University, Via Rita Levi Montalcini 4, Pieve Emanuele, 20072 Milan, Italy; 3Endocrinology, Diabetology and Medical Andrology Unit, IRCCS Humanitas Research Hospital, Via Manzoni 56, Rozzano, 20089 Milan, Italy; 4Endocrinologia e Malattie Metaboliche, AO SS Antonio e Biagio e Cesare Arrigo, Via Venezia, 16, 15121 Alessandria, Italy; 5Department of Experimental Medicine, “Sapienza” University of Rome, 00161 Rome, Italy; 6Operative Unit of Endocrinology, Department of Clinical Medicine and Surgery, Andrology and Diabetology, Federico II University of Naples, 80131 Naples, Italy; 7Servizio di Endocrinologia Oncologica, European Institute of Oncology, IEO, IRCCS, Via Ripamonti 435, 20141 Milan, Italy; 8Endocrinology Unit, Department of Clinical Medicine and Surgery, University Federico II, 80126 Naples, Italy; 9UNESCO “Education for Health and Sustainable Development”, Federico II University, 80131 Naples, Italy; 10Endocrinology Unit, Department of Clinical and Molecular Medicine, Sant’Andrea Hospital, ENETS Center of Excellence, Sapienza University of Rome, 00185 Rome, Italy

**Keywords:** gastric NET, type 3, neuroendocrine tumors, stomach

## Abstract

**Simple Summary:**

Gastric well-differentiated neuroendocrine tumors (g-NETs) occur in 5–15% of cases of neuroendocrine neoplasms in the gastrointestinal tract, and they mostly present quite indolent biological behavior. Furthermore, among the g-NETs, the type 3 versions represent the most aggressive subcategory, with the highest incidence of lymph-vascular infiltration and metastases at diagnosis. For these reasons, a more interventional approach to the management of type 3 g-NETs has been generally shared by the scientific community. However, a case-by-case discussion amongst a multidisciplinary NET-dedicated team revealed that partial or complete gastrectomy with lymphadenectomy had become an out-of-proportion indication within such cases regarding small and/or low-grade lesions. The aim of this systematic review was to assess the real-life approaches to type 3 g-NETs and identify some of the prognostic factors that might impact management choices. We found that size and gastric wall infiltration and grading may represent three factors that should be taken into account to better manage type 3 g-NETs.

**Abstract:**

Purpose: to collect data from real-life experiences of the management of type 3 g-NETs and identify possible prognostic factors that may impact the decision-making process. Methods: We performed a systematic review of the literature on type 3 g-NET management using the PubMed, MEDLINE, and Embase databases. We included cohort studies, case series, and case reports written in the English language. Results: We selected 31 out of 556 articles from between 2001 and 2022. In 2 out of the 31 studies, a 10 mm and 20 mm cut-off size were respectively associated with a higher risk of gastric wall infiltration and/or lymph node and distant metastasis at diagnosis. The selected studies reported a higher risk of lymph node or distant metastasis at diagnosis in the case of muscularis propria infiltration or beyond, irrespective of the dimensions or grading. From these findings, size, grading, and gastric wall infiltration seem to be the most relevant factors in management staff making choices and prognoses of type 3 g-NET patients. We produced a hypothetical flowchart for a standardized approach to these rare diseases. Conclusion: Further prospective analyses are needed to validate the prognostic impact of the use of size, grading, and gastric wall infiltration as prognostic factors in the management of type 3 g-NETs.

## 1. Introduction

Well-differentiated gastric neuroendocrine tumors (NETs) of the stomach are rare tumors and represent 5–15% of whole gastrointestinal (GI) NETs [1]. The incidence of these tumors has increased in recent years due to the widespread use of endoscopy [2].

Gastric NETs (g-NETs) originate from histamine-secreting enterochromaffin-like cells (ECL) within the mucosa of the stomach. As with other gastro-enteropancreatic (GEP) NETs (poorly differentiated morphology apart), g-NETs are classified according to the 2019 World Health Organization (WHO) Classification as being grade (G) 1, G2, and G3 based on the proliferation index by Ki-67 (<3%, between 3 and 20%, and >20% respectively) [3]. Furthermore, g-NETs also undergo the Rindi et al. classification that subdivides the categories according to clinical and pathological features and biological behavior [4,5].

The most common and the oldest subcategories are represented by types 1, 2, and 3. Type 1 g-NETs are typically associated with autoimmune atrophic gastritis and elevated serum gastrin and achlorhydria. The lesions are often multiple in number and are characterized by indolent behavior (metastases in 0–2% of cases) [6,7]. Type 2 is generally detected within multiple endocrine neoplasia (MEN) type 1 syndrome. They are characterized by hypergastrinemia and high gastric acid secretion conditioned by a gastrinoma. Additionally, type 2 g-NETs have quite indolent behavior but present 12–30% of the risk of distant metastases at diagnosis [8]. Type 3 g-NETs represent the most aggressive subcategory and are not associated with hypergastrinemia, atrophic gastritis, or MEN1 syndrome. Type 3 g-NETs typically occur as a single, sporadic lesion with a larger size and present more aggressive biological behavior compared with types 1 and 2, which is represented by a deeper level of stomach wall invasion, lymph-vascular infiltration, and a higher risk of metastasis at diagnosis [4].

For all these reasons, the European and international guidelines agree on the more aggressive management of type 3 g-NETs. According to the NANETS guidelines, the type 3 subcategory is sufficient for the indication of partial gastrectomy and lymphadenectomy, irrespective of the grade or dimension [9]. The European Neuroendocrine Tumors Society (ENETS) and National Comprehensive Cancer Network (NCCN) guidelines keep the possibility of an endoscopic approach for small and superficial lesions [10]. However, both of the latter suggest endoscopic resection without distinguishing among the different gastric types but using only the dimensional criteria for the patient’s selection [11,12].

This systematic review aimed to collect all the available data on type 3 g-NETs to identify real-life managerial experiences and eventually provide a flowchart for management regarding these rare and aggressive diseases.

## 2. Materials and Methods

On behalf of the NIKE (Neuroendocrine tumors, Innovation in Knowledge and Education) group in Italy, we performed a systematic review of the literature according to Cochrane collaboration and the “Preferred Reporting Items for Systematic Reviews and Meta-Analyses” (PRISMA) statement methodology [13], using the PubMed, MEDLINE, and Embase databases (see Appendix A). Other relevant studies were identified through the references list of included papers. We included all relevant clinical articles (cohort studies, case series, and case reports) written in the English language, with those from inception to 2022 included. We selected only those studies focusing on or containing patients with type 3 g-NETs (defined as sporadic g-NET) not associated with hypergastrinemia or history of autoimmune atrophic gastritis, hypochlorhydria, gastrinomas, Zollinger-Ellison syndrome, or MEN-1. Studies were excluded if they were not written in English, did not represent a clinical study (e.g., reviews), or were not related to type 3 gastric NETs. Articles were screened by reading the title and abstract, and the full text of the eligible studies was obtained.

For every selected article, we extracted (through a standardized form) the data regarding year of publication, number of participants, demographic characteristics of patients, clinic-pathological characteristics of gastric NETs, and information concerning disease management and treatments. When available, we also gathered data regarding survival and mortality.

Data are expressed as mean and standard deviation or median and interquartile range (IQR), as appropriate. Categorical variables are expressed by the absolute number and percentage. All statistical analyses were performed with Microsoft Excel (Microsoft Corporation, Redmond, WA, USA) and MedCalc Statistical Software version 20.211 (MedCalc Software Ltd., Ostend, Belgium).

## 3. Results

Among the 556 identified articles in the 21-year period between 2001 and 2022, we selected 31 studies for the final analysis. Figure 1 represents the PRISMA flowchart of the literature research.

Among the selected studies, 7 are case reports, 20 are retrospective cohort studies (9 multicentric), and 4 are prospective cohort studies (3 multicentric). A total of 776 patients with type III g-NETs were included in this systemic review. Their characteristics are reported in Table 1 and Table 2.

The median age of diagnosis was 58 years (range 34–71), and males represented 59.3% of the population. In 21 studies, the origin of the tumor site was predominantly the gastric corpus/fundus (83.3%) [14,15,16,17,18,19,20,21,22,23,24,25,26,27,28,29,30,31,32,33,34], and prevalent polypoid morphology from endoscopy was identified in 13 out of the 31 studies [15,16,17,20,21,22,23,25,26,28,29,30,31]. A total of 17 out of the 31 studies included tumor depth invasion data, resulting in an invasion that included the submucosa in most cases [15,16,19,20,21,22,23,24,25,28,30,32,34,35,36,37,38]. A total of 26 out of the 31 papers reported dimensional data, and the median size was 16.5 mm (8–62.5 mm) [14,15,16,17,18,19,20,21,23,24,26,28,31,32,34,36,37,38,39,40,41,42,43,44,45]. Six out of thirty-one authors reported a cut-off size of >10 mm for low tumor-related death [16,17,18,19,21,39], while 7 out of the 31 papers reported a larger cut-off size of >20 mm for an increased risk of gastric wall infiltration, metastasis at diagnosis, and relapse (*p* < 0.001) [14,22,23,24,26,43,44]. A total of 3 out of the 31 authors showed a higher risk of gastric wall infiltration (and then lymph node and distant metastasis at diagnosis) with a cut-off of >17 and >19 mm [30,37,38], while two authors shared a stricter cut-off of <5 mm for sharing an indication of endoscopic management [37,39]. The data on lymphatic and vascular invasion were available in less than 50% of the selected articles. A total of 13 studies reported data on gastric wall infiltration as being related to the patient’s outcome; muscularis propria involvement was associated with a higher risk of lymph node or distant metastasis, irrespective of the size and grading [15,19,20,23,24,26,30,31,32,33,37,43,44]. In particular, Sato et al. described a G1 8 mm lesion with a lymph node metastasis at diagnosis due to muscularis propria infiltration [15], while Min et al. showed no lymph nodes or distant metastases for all type 3 g-NETs (even G3) confined in the submucosa [23].

Grading was defined in 25 studies (n = 648): G1 type 3 g-NETs = 298 (46%), while G2/G3 = 350 (54%) [14,15,16,17,18,19,20,21,22,23,24,25,26,27,29,30,31,33,35,36,37,39,41,43,46], with a mean Ki-67 of 18.2% (reported in 15 out of the 31 studies) [15,16,17,21,22,24,25,26,29,33,36,37,39,41,46]. Lymph node and distant metastasis at diagnosis were reported in 22 studies, with 27.9% of the patients having lymph node metastasis [14,15,16,19,20,21,22,23,24,25,26,27,28,34,35,36,37,38,42,43,44,46], while 25.2% had distant metastasis (available data in 26 studies) [14,15,16,17,20,21,22,23,24,25,26,27,28,29,31,33,34,35,36,37,38,41,42,43,44,46]. More than half of the patients with type 3 g-NETs received surgical treatment (total gastrectomy with Roux-en-Y reconstruction). Among the 43.8% of patients that underwent endoscopic management, 6% received subsequent surgical treatment [14,15,16,18,19,21,22,23,24,25,39,41,42,43,46]. A total of 11 studies reported data on disease relapse after radical treatment (47/444 patients), but no homogeneous data could be identified for the site of recurrence (locoregional end/or distant), the size of local recurrences, the time to recurrence from local treatment, and the recurrence grade [18,19,20,23,28,36,37,39,41,44,46]. The number of patients treated with systemic therapy (chemotherapy, somatostin analogs, and radionuclide therapy) totaled 84 according to 13 studies, and chemotherapy was the main resulting systemic treatment received [18,21,22,27,29,37,38,39,41,42,43,44,46].

Eleven of the papers reported survival rate data. Six studies evaluated the 5-year (y) overall survival (OS), which resulted in between 100% and 54.5% (39.5–69.5) [19,23,34,39,41,43].

The highest 5y OS was observed in the studies by Hirasawa et al.: 100% in 63 out of 144 patients treated with endoscopic resection. The sample was entirely composed by G1 and G2 type 3 g-NETs, with no involvement of the submucosa in 88.2% of cases [19]. Additionally, the study by Min et al. reported a high median 5y OS (96%) in a sample composed by G1/2/3 confined to the submucosa in almost 94% of cases. The median size was 15 mm, and 22 out of 32 patients received endoscopic management for type 3 g-NETs [23].

Hanna et al. reported a 5y OS of 81% for a sample composed of G1/2/3 type 3 g-NETs. However, the 5y OS referred to the whole sample for whom 13 out of the 47 patients underwent endoscopic management [39]. Additionally, Panzuto et al. reported a case series of G1/2/3 type 3 g-NETs with a similar 5y OS (76.2%). The median size was 15 mm, and 16 out of 19 of the patients underwent endoscopic resection [41]. The survival reported by Manfredi and Schindl was appreciably lower (5y OS of 63.2% and 54.5%, respectively). Both the authors reported a larger median size, and the case series by Schindl et al. (despite the fact that the grading was not specified) showed a more advanced stage [34,43].

The studies by Jiao and Li YL reported a quite different 3y OS: 29.9% and 75%, respectively [21,22], but the first analysis reported a higher percentage of NET G3 (16/25) and a higher median size (35 mm) compared to the Li YL report and the other previous studies. Furthermore, Endo et al. reported a survival rate of 66% at 53 months follow-up for a sample of G1/2/3 type 3 g-NETs with a higher median size (55 mm) and proliferation index (51%) compared to the studies above [37].

Postelwait et al. reported a 75% 3y-disease-specific survival (DSS). The population was composed of 10 patients, with a median size of 32 mm (3–61), and the grading was not specified. Five of the patients had a stage of ≥T3, and only two patients received endoscopic management [44].

Lastly, Louthan et al. reported the poorest survival (a median of 3 months) for a cohort of seven type 3 g-NET patients. The reason for these findings was that the sample was composed of six (out of the seven) patients with metastatic disease at diagnosis, and four of these were G3 [27].

## 4. Discussion

To our knowledge, this systematic review is the first comprehensive analysis of all the studies on type 3 g-NETs from the last 20 years. Among the different subcategories, type 3 represents 15–25% [10] of g-NETs, with an incidence of 6.15 million people [47]. Due to rarity, the lack of a specific conditioning cause, and the natural trend of more aggressive-type behavior, type 3 g-NETs are mostly associated with a poor prognosis [48]. For these reasons, the European and international guidelines agree on more aggressive therapeutic management for all localized stage types 3 g-NETs. An endoscopic approach seems to be suggested only for small and superficial lesions, using only the dimensional criteria for the patient’s selection [10,48]. Chen et al. supported this indication and reported their experience with endoscopic resection in 10 type 3 g-NETs. In this study, management was based on dimensional criteria (the median size of all type 3 g-NETs was almost 16.5 mm, with a range of 8–30 mm) and the infiltration depth of the gastric wall (none exceed the submucosal layer) [30].

According to the results of our analysis, size, grading, and gastric wall infiltration seem to be the most important decision points in managerial choices and are the most relevant factors that impact the prognosis of type 3 g-NET patients.

When considering only the dimensional criteria, most of the reported studies agree on a quite long disease-free survival for patients with <10 mm lesions treated with endoscopic resection.

When adding gastric wall infiltration to the dimensional parameters, some of the authors argue about the indication of endoscopic management for lesions of up to 20 mm in diameter, irrespective of the grade, as long as the type 3 NET result is confined within the submucosal area. Involvement beyond the submucosa is generally (but not always) related to larger lesions and might, indeed, be associated with a risk of lymph-vascular infiltration, positive margins after endoscopic dissection, lymph node positivity, and metastatic disease at staging. Obviously, larger lesions at diagnosis and greater gastric wall involvement might be related to higher-grade lesions due to more aggressive behavior, but according to our observations, surgical management should be discussed case by case. In any case, while endoscopic management may be more easily accepted by the scientific community for small G1/2 type 3 g-NETs confined within the submucosa, as regards a partial/total gastrectomy, for NETs G3 with the same characteristics, it may be more difficult. NETs G3 are indeed the subcategory with the worst prognosis among the well-differentiated NENs, and physicians may not be so confident about their biological behavior. However, we believe that the impact of gastrectomy on patients’ quality of life and the risk of second tumors from the surgery site represent issues that should be well-considered in the therapeutical planning for type 3 g-NETs.

According to the proliferation index, Li YL et al. suggested endoscopic treatment only for patients with <10% G2 g-NETs, while Hirasawa et al. directly proposed surgery for all G2 g-NET cases. However, these cut-offs were not confirmed by the other studies, in which, in case of <17 mm size lesions confined within the submucosa, nor local or distant relapse, neither metastatic disease at diagnosis have been reported, irrespective of the grade.

Based on our observations, the dimensional criteria should be the first parameter to consider after the endoscopic biopsy of a type 3 g-NET and adequate staging (according to the grade). Correlating the outcomes and g-NET features reported by the selected studies, we measured a cut-off of 17 mm in diameter from the median of the tumor sizes observed.

The second parameter that we considered was the state of gastric wall infiltration. From the results of our analysis, muscularis propria infiltration (or beyond) was directly related to lymph-vascular infiltration, lymph nodes, and distant metastases. Although some studies showed a correlation of 67–100% between muscularis propria infiltration risk and a diameter of >20 mm, some others reported cases of <20 mm NETs with a deeper infiltration than expected. An example is the case by Sato et al., who reported an 8 mm G1 g-NET with a muscle layer adhesion; further radical surgery after the endoscopic dissection showed two lymph nodal metastases.

Therefore, for small lesions, the assessment of muscularis propria infiltration may contribute to an indication for surgery. Conversely, due to the higher correlation with lymph vascular infiltration, lymph nodes, and distant metastases, a size of ≥17 mm may be sufficient to indicate surgery if a metastatic disease has been excluded at the staging.

In the case of metastatic disease, indications will follow international guidelines, which are based on the grade, the radiologic characteristics of the disease, the tumor burden, and the clinical features of the patients.

According to these observations, we reported a hypothetical flowchart for management staff in the case of an endoscopic detection of a type 3 g-NET based on T stage (size and gastric wall infiltration) and grade (Figure 2).

Our systematic review clearly presents some limitations that prevent us from adopting this flowchart without prospective validation. On the one hand, the selected studies belong to different eras in terms of NEN classification, knowledge of NEN features, which have been improved over the years, and an awareness of the importance of centralizing these rare diseases in referral centers. On the other, the sample analyzed is also heterogeneous in terms of race, surgery approach (total vs. partial gastrectomy), and time of follow-up. However, the strength of the present study is represented by the sample size, which is uncommon for these rare diseases, the strict selection criteria of the studies and the cases within them, and the result of a shareable management flowchart that comes from real-life experiences.

## 5. Conclusions

According to the present systematic review on type 3 g-NETs, real-life managerial experiences seem to be influenced by size, grading, and gastric wall infiltration, which all have a certain prognostic impact on the outcomes of these patients. Prospective analyses are needed to validate the usefulness of our management flowchart, which might aid the standardization of the therapeutic approach to type 3 g-NET patients.

## Figures and Tables

**Figure 1 cancers-15-02202-f001:**
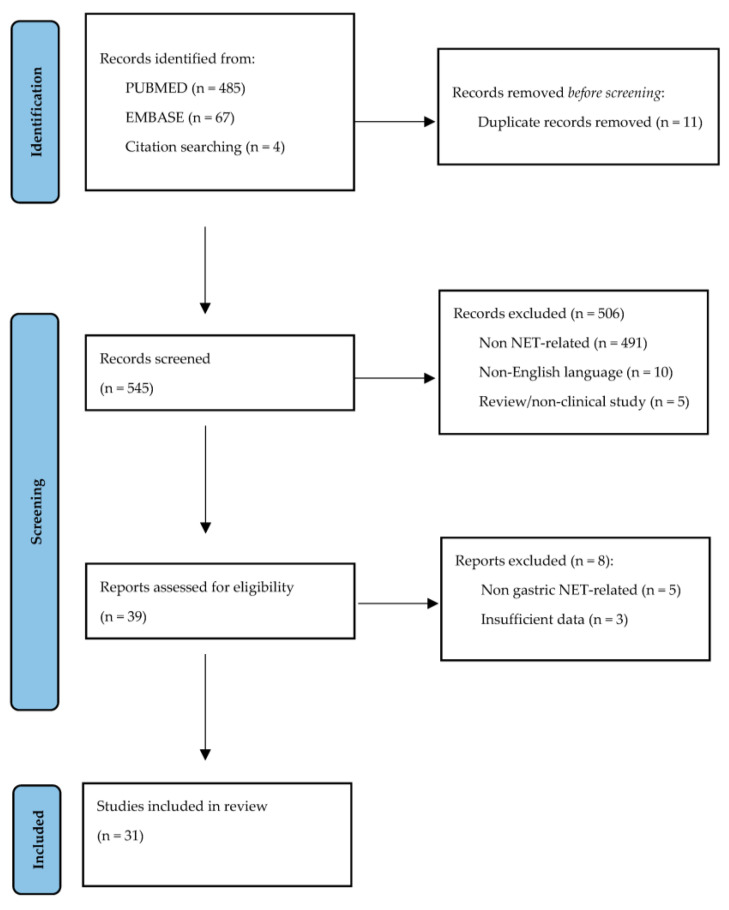
Flowchart depicting the literature search for the systematic review, in accordance with the PRISMA statement.

**Figure 2 cancers-15-02202-f002:**
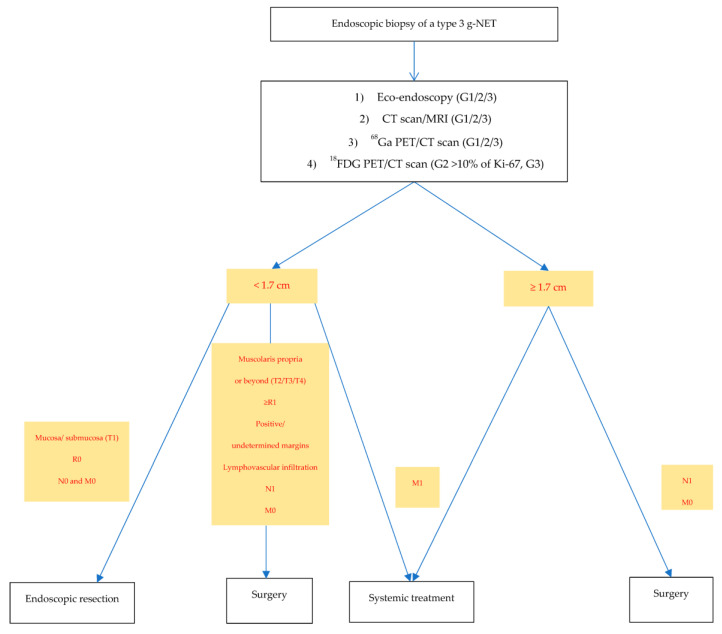
Flowchart management in the case of endoscopic type 3 g-NET detection. CT: computer tomography, Ga: gallium, PET: Positron Emission Tomography, FDG: fluorodeoxyglucose, G: grade, R0: absence of microscopic residual of disease, N0: no positive lymph nodes at the radiologic imaging, M0: no distant metastasis at the radiologic imaging, N1: positive lymph nodes at the radiologic imaging, M1: distant metastasis, R1: microscopic residual of disease.

**Table 1 cancers-15-02202-t001:** Demographic, pathological, and staging characteristics of patients with type 3 g-NETs.

Year	First Author	Patients (n)	Sex (M = Male; F= Female)	Mean Age (y)	Site (A = Antrum; C = Corpus; F = Fundus)	Endoscopic Appearance (P = Polyp; U = Ulcer)	Depth of Invasion (M = Mucosa; MP = Muscularis Propria; S = Serosa; SM = Submucosa; SS= Subserosa)	Grading	ki-67 (%)	Stage T	Stage N	Stage M	Median Size (mm)
2022	Sekar A.	23	16M-7F	47.4	NA	NA	1MP-2S-20 NA	G1 (n = 11), G2 (n = 5), G3 (n = 7)	NA	NA	N1 (n = 9)	M1 (n = 6)	NA
2022	Kurtulan O.	14	10M-4F	64.5	NA	NA	NA	G2 (n = 3), G3 (n = 11)	NA	T3-4 (n = 14)	N1 (n = 9)	M1 (n = 5)	62.5
2022	Sato A.	1	F	42	C	P	1SM	G1	2	T1	N1	M0	8
2022	Zhu C.	1	M	34	C	P	1M	G3	80	T1	N0	M0	4
2022	Boeriu A.	1	M	56	A	P	NA	G2	5	T2	NA	M1	20
2021	Hanna A.	47	20M-27F	56.6	NA	NA	NA	G1 (n = 22), G2/3 (n = 25)	16	T1 (n = 29), T2 (n = 18)	NA	NA	10
2021	Exarchou K.	45	24M-21F	56	3A-36C-6F	NA	NA	G1 (n = 22), G2/3 (n = 22) NA (n = 1)	NA	NA	NA	NA	12
2021	Hirasawa T.	144	88M-56F	62	20A-81C-43F	NA	13 M-114SM-11 MM-5 SS-1 S	G1 (n = 90), G2 (n = 54)	NA	T1 (n = 127), T2 (n = 11), T3 (n=5), T4 (n=1)	N1 (n = 15)	NA	8
2020	Trinh V.Q.	66	33 M-33F	60	6A-60C/F	42P-6U-18 others	23 M-31 SM-5 MP-1 SS-6 S	G1 (n = 34), G2 (n = 28), G3 (n = 4)	NA	T1 (n = 38), T2 (n = 21), T3 (n = 1), T4 (n = 6)	N1 (n = 12)	M1 (n = 4)	16
2020	Jiao X.	25	20M-5F	60	3A-3C-10F-9 others	14P-6U-	2 M/SM-10 MP/S-15 NA	G1 (n = 2), G2 (n = 7), G3 (n = 16)	34	T1 (n = 2), T2 (n = 4), T3 (n = 8) NA (n = 11)	N1 (n = 4)	M1 (n = 13)	35
2020	Li Y.L.	77	34M-43F	48	3A-64C/F-10 others	45P-17U	22M/SM-5 MP-7 beyond MP	G1 (n = 37), G2 (n = 31), G3 (n = 9)	3	NA	N1 (n = 10)	M1 (n = 24)	15
2019	Panzuto F.	19	6M-13F	59	NA	NA	NA	G1 (n = 8), G2 (n = 10), G3 (n = 1)	3	NA	NA	M1 (n = 7)	15
2018	Crown A.	37	NA	NA	NA	NA	NA	NA	NA	T1 (n = 12), T2 (n = 14), T3 (n = 4), NA (n = 7)	N1 (n = 9)	M1 (n = 5)	28
2018	Min B.H.	32	23M-9F	52	7A-25C/F	32P	31M/SM-1MP	G1 (n = 25), G2 (n = 5), G3 (n = 2)	NA	NA	N1 (n = 2)	M1 (n = 1)	9.5
2018	Vanoli A.	34	24M-10F	59	2A-32C/F	NA	13SM-21MP	G1 (n = 15), G2 (n = 10), G3 (n = 9)	5% (median G2) 39% (median G3)	NA	N1 (n = 11)	M1 (n = 10)	20
2017	Manfredi S.	52	33M-19F	58	NA	NA	NA	G1 (n = 10), G2 (n = 26), NA (n = 16)	NA	NA	N1 (n = 9)	M1 (n = 27)	20
2017	Schmidt D.	1	M	68	NA	NA	NA	G2	5	T4	N1	M1	NA
2016	Kawasaki K.	1	M	69	C	P	SM	G2	4,8	T1	N0	M0	NA
2016	Lee H.E.	22	11M-11F	6.2	NA	NA	13M/SM-2MP-7 SS/S	G1 (n = 3), G2 (n = 15), G3 (n = 4)	13	T1 (n = 13), T2 (n = 2), T3 (n = 6), T4 (n = 1)	N1 (n = 7)	M1 (n = 3)	23
2016	Postlewait L.M.	10	5M-5F	58.8	NA	NA	NA	NA	NA	T1 (n = 2), T2 (n = 1), T3 (n = 2), T4 (n = 3) NA (n = 2)	N1 (n = 5)	M1 (n = 4)	32
2014	Cavallaro A.	1	M	66	C	SM	NA	G1	<2%	T2	N1	M0	8
2014	Louthan O.	7	5M-2F	66	4F-3C	NA	NA	G2 (n = 3), G3 (n = 4)	NA	NA	N1 (n = 2)	M1 (n = 6)	NA
2013	Kwon Y.H.	50	28M-22F	58.6	4A-38C-8F-7 other	48P	49M/SM-1MP	NA	NA	T1 (n = 49), T2 (n = 1)	N0 (n = 50)	M0 (n = 50)	≤10 (n = 33) >10 (n = 17)
2013	Bariani G.M.	1	M	64	C	U	NA	G1	1	NA	NA	M1	NA
2012	Chen W.	10	3M-7F	53.5	1A-7C-1F-1 other	1U-9 other	10SM	G1 (n = 7), G2 (n = 3)	NA	NA	NA	NA	16.5
2012	Endo S.	8	5M-3F	67	NA	NA	1M-3SM-3S-2NA	G1 (n = 3), G2 (n = 2), G3 (n = 3)	3.5	Tis (n = 1), T1 (n = 2), T2 (n = 2), T3 (n = 3)	N1 (n = 2)	M1 (n = 1)	20
2012	Li Q.L.	8	3M-5F	56	5C-1A-1F-1 other	8 SM	NA	G1 (n = 6), G2 (n = 2)	NA	NA	NA	M0	16.5
2010	Kim B.S.	16	10M-6F	51.1	8F-6C-2A	NA	13SM-1MP-1NA	NA	NA	NA	NA	NA	11.7
2007	Safatle-Ribeiro A.V.	8	4M-4F	60.5	5C-3A	NA	NA	G2 (n = 1), G3 (n = 7)	45	NA	NA	M1 (n = 5)	NA
2005	Borch K.	4	1M-3F	71	NA	NA	2MP-2S	NA	NA	T2 (n = 2), T4 (n = 2)	N0 (n = 4)	M1 (n = 3)	19
2001	Schindl M.	11	NA	NA	5A-6C/F	NA	2M/SM-2MP-2 SS/S	NA	NA	T1 (n = 2), T2 (n = 2), T4 (n = 7)	N1 (n = 9)	M1 (n = 4)	41.6

NA: not available, G: grade.

**Table 2 cancers-15-02202-t002:** Treatment approaches, follow-up, and survival data of patients with type 3 g-NETs.

Year	First Author	Patients (n)	Local Treatment (E = Endoscopy; S = Surgery)	Systemic Treatment (C = Chemotherapy; R = Radionuclide Therapy; S = Somatostatin Analogues)	Median Follow-Up (Months)	Relapse (D = Distant; L = Local; LR = Loco-Regional)	Site of Distant Relapse	Survival	Deceased due to NET
2022	Sekar A.	23	NA	NA	NA	NA	NA	NA	NA
2022	Kurtulan O.	14	14S	NA	NA	NA	NA	NA	NA
2022	Sato A.	1	E-S	NA	NA	NA	NA	NA	NA
2022	Zhu C.	1	E	No	12	No	No	NA	NA
2022	Boeriu A.	1	No	NA	NA	NA	NA	NA	NA
2021	Hanna A.	47	13E-26S-8NA	NA	62.7	11LR/D	NA	5y-OS 81%	7
2021	Exarchou K.	45	16E-26S-3NA	NA	56	2	Liver	NA	0
2021	Hirasawa T.	144	63E (15 with subsequent S)-81S	No	32 (endoscopic group) 51 (surgery group)	7D (2 in endoscopic group, 5 in surgery group)	Liver (n = 6), bone (n = 2), LN (n = 1)	OS 100% (endoscopic group) OS 91.8% (surgery group)	5
2020	Trinh V.Q.	66	NA	NA	49	5L-3D	NA	NA	3
2020	Jiao X.	25	2E-11S-12NA	2S-14C-9NA	NA	NA	NA	3y-OS 29.9%	NA
2020	Li Y.L.	77	33E-17S-27NA	6S-21C-50NA	35	NA	NA	3y-TSS 75%	NA
2019	Panzuto F.	19	6E-9S-4NA	3S-2R-14NA	24	3	NA	5y-OS 75%	2
2018	Crown A.	37	8E-29S	1C-36NA	NA	NA	NA	NA	NA
2018	Min B.H.	32	22E-10S	1C-31NA	59	1LR-1D	Liver (n = 1)	5y-OS 96%	1
2018	Vanoli A.	34	8E-26S	NA	93	NA	NA	NA	13
2017	Manfredi S.	52	15E-17S-20NA	3S-2C-47NA	24	NA	NA	5y-OS 63.2%	NA
2017	Schmidt D.	1	S	S-R-C	79	L-D	Liver, bone, LN	NA	No
2016	Kawasaki K.	1	E	No	NA	NA	NA	NA	NA
2016	Lee H.E.	22	NA	NA	14	1LR-3D	Liver (n = 3), omentum (n = 1)	NA	1
2016	Postlewait L.M.	10	2E-8S	NA	30.7	1LR-1D	Liver	3y-TSS 75%	2
2014	Cavallaro A.	1	S	No	NA	NA	NA	NA	
2014	Louthan O.	7	1S	3S-3C-1NA	46.8	NA	NA	Median OS 3 months	6
2013	Kwon Y.H.	50	50E-39S (previous endosocpic treatment)	No	46	4	NA	NA	3
2013	Bariani G.M.	1	No	S-C	34	NA	NA	NA	Yes
2012	Chen W.	10	10E-1S (previous endoscopic treatment)	No	27.5	No	NA	OS 100%	No
2012	Endo S.	8	1E-6S-1NA	3C-5 NA	42.5	1	Liver	3.5 ys survival 87.5%	1
2012	Li Q.L.	8	8E-1S (previous endoscopic treatment)	No	27	No	NA	NA	
2010	Kim B.S.	16	7E-7S-2NA	NA	68	No	NA	OS 100%	No
2007	Safatle-Ribeiro A.V.	8	5S-3NA	NA	12	NA	NA	OS 28.6%	5
2005	Borch K.	4	4S	4C	95	NA	NA	NA	1
2001	Schindl M.	11	2E-4S	No	19.1	1	NA	5y-OS estimation 54.5%	7

LN, lymphnodes; NA, not available; OS, overall survival; TSS, tumor-specific survival; ys, years.

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
