# Peer review of "Gastric Neuroendocrine Tumors (g-NETs): A Systematic Review of the Management and Outcomes of Type 3 g-NETs"

_cancers, 2023, doi:10.3390/cancers15082202_

Round 1

Reviewer 1 Report

The manuscript consists in a systematic review of studies reporting the management and outcomes of type 3 gastric NENs, with a special focus on the use of endoscopic resection. Following PRISMA guidelines, the authors identified 31 studies, all of them from 2001 onwards. 776 patients were considered for the analysis. Finally, a management algorithm was proposed.

This is an interesting work, which analyzes a relevant population with a diagnosis of neuroendocrine tumor and suggests a rational approach in a disease setting where surgical management is still debated.

 Major remarks:

·       It is not clear how the tumor size cut-off proposed in the management algorithm (17 mm) was obtained. In the manuscript it is reported that “correlating the outcomes and gastric NETs features reported by the selected studies, we measured a median cut-off of 17 mm in diameter”. I would request the authors to elucidate whether this cut-off is the approximation of the median tumor size observed (16.5 mm) or it derives from an inferential analysis (in this case, please specify the type of analysis).

·       No statistical inference appears to be reported in the Results. Therefore, the description of statistical tests in the Methods should be removed.

·       Proof-reading and extensive English refining are required, e.g.:
- Lines 39-40: “management has been generally shared” -> proposed?;
- Line 41: “partial or complete gastrectomy with lymphadenectomy has become an indication out of proportion” – in what sense?
- Line 95: “rare and aggressive rare diseases” (repetition)
- Line 204: “(n= and not n: 648)”

- Line 207: “The lymph node and distant metastasis at diagnosis was reported in 22/31 studies” -> metastases… were;
- Line 215 :  “size of recurrences primaries” - What does it mean?

 Minor remarks:

·       Table 1 and 2: punctuation needs to be aligned and patients’ distribution in columns need to be checked (i.e.: Kurtulan: n= 14 patients, 37 with corpus neoplasia ???).

·       IQT intervals and median ranges are reported in an unconventional way and are not consistent in the manuscript:
- Range of median size should be reported from low to high and not vice versa (i.e., lines 189 and 260).
- Line 221: “the 5 years (y)-overall survival (OS) that resulted between 100% and 54.5±15%”. Please report a conventional range removing ±.

·       - Line 243: “a median size of 32±29mm” -> Please specify median ranges instead of standard deviation.

Author Response

Point by point answers

Reviewer 1

We’d like to thank the reviewer for the observations.

Major remarks:

  • It is not clear how the tumor size cut-off proposed in the management algorithm (17 mm) was obtained. In the manuscript it is reported that “correlating the outcomes and gastric NETs features reported by the selected studies, we measured a median cut-off of 17 mm in diameter”. I would request the authors to elucidate whether this cut-off is the approximation of the median tumor size observed (16.5 mm) or it derives from an inferential analysis (in this case, please specify the type of analysis).

We thank the reviewer for the observation. We edit the sentence specifying the analysis.

  • No statistical inference appears to be reported in the Results. Therefore, the description of statistical tests in the Methods should be removed.

We’d like to thank the reviewer for this observation. We remove the sentence from the methods.

  • Proof-reading and extensive English refining are required, e.g.:

- Lines 39-40: “management has been generally shared” -> proposed?;

We mean “shared within the scientific community”. We edit accordingly.

- Line 41: “partial or complete gastrectomy with lymphadenectomy has become an indication out of proportion” – in what sense?

We’d like to thank the reviewer for the question because we were afraid the sentence wasn’t so clear. We edit the sentence accordingly.

- Line 95: “rare and aggressive rare diseases” (repetition)

Thank you for your report. We erase the second “rare”

- Line 204: “(n= and not n: 648)”

Thank to the reviewer for the observation. We edit the sentence.

- Line 207: “The lymph node and distant metastasis at diagnosis was reported in 22/31 studies” -> metastases… were;

Thank to the reviewer for the observation. We edit the sentence.

- Line 215 :  “size of recurrences primaries” - What does it mean?

Thank to the reviewer to the observation. We edit the sentence accordingly

 Minor remarks:

  • Table 1 and 2: punctuation needs to be aligned and patients’ distribution in columns need to be checked (i.e.: Kurtulan: n= 14 patients, 37 with corpus neoplasia ???).

We’d like to thank the reviewer for the observation. We edit the tables accordingly

  • IQT intervals and median ranges are reported in an unconventional way and are not consistent in the manuscript:

- Range of median size should be reported from low to high and not vice versa (i.e., lines 189 and 260).

- Line 221: “the 5 years (y)-overall survival (OS) that resulted between 100% and 54.5±15%”. Please report a conventional range removing ±.

  • - Line 243: “a median size of 32±29mm” -> Please specify median ranges instead of standard deviation.

We thank the reviewer. We edit the text accordingly

Reviewer 2 Report

This paper concerning type 3 gastric neuroendocrine neoplasms is a well written systematic review on this unusual topic dealing with rare gastric NENs. The paper is limited to data coming from English literature review, but the authors (from nine well-known different Departments of Gastroenterology, Endocrinology and Oncology in Italy) do not report even a single personal case on this subject that would have provide a “novel insight on this entity”.

-          In the “Methods” section add what NIKE group means (‘NIKE’ = Neuroendocrine tumors Innovation Knowledge and Education). Please, provide the reasons why the Authors did not include in this review their collective experience on type 3 gastric NENs, for example in Tables 1 and 2.

-          As shown in Table 1, median tumor size was reported in 25 out of 31 studies, depth of gastric wall invasion (T stage) was reported in 16 out of 31 studies, and tumor grade (G) was reported in 25 out of 31 studies. As very well known in the current literature, grade and stage are the most important prognostic factors for NENs. Since from this review tumor grade emerged as a prognostic factor, I recommend to include it (all grades) in the flowchart management (Figure 2).

-          Grade 3 is reported as having the worse prognosis (OS 29.9% at 3 years, as reported by Jiao et al. [21]). Therefore, in the right part of the flowchart even G3 in addition to size > 1.7 cm should undergo surgery. Similarly, I would recommend to add how to manage G1 and G2 gastric NENs (in the left part of the flowchart?), depending on the findings from the Authors’ analysis of the current literature.

-          Concerning Figure 2, I recommend to include in the flowchart what to do if histology after endoscopic treatment shows undetermined margins.

Author Response

Point by point answers

We'd like to thank the reviewer for the observations. We answered to the questions of the reviewer and edited the text accordingly. 

This paper concerning type 3 gastric neuroendocrine neoplasms is a well written systematic review on this unusual topic dealing with rare gastric NENs. The paper is limited to data coming from English literature review, but the authors (from nine well-known different Departments of Gastroenterology, Endocrinology and Oncology in Italy) do not report even a single personal case on this subject that would have provide a “novel insight on this entity”.

-          In the “Methods” section add what NIKE group means (‘NIKE’ = Neuroendocrine tumors Innovation Knowledge and Education). Please, provide the reasons why the Authors did not include in this review their collective experience on type 3 gastric NENs, for example in Tables 1 and 2.

We’d like to thank the reviewer for the observation. We did not include our experience because we aimed to perform a systematic review of the crude literature data.

-          As shown in Table 1, median tumor size was reported in 25 out of 31 studies, depth of gastric wall invasion (T stage) was reported in 16 out of 31 studies, and tumor grade (G) was reported in 25 out of 31 studies. As very well known in the current literature, grade and stage are the most important prognostic factors for NENs. Since from this review tumor grade emerged as a prognostic factor, I recommend to include it (all grades) in the flowchart management (Figure 2).

Thank the reviewer for the observations. We edit the figure according with the suggestions.

-          Grade 3 is reported as having the worse prognosis (OS 29.9% at 3 years, as reported by Jiao et al. [21]). Therefore, in the right part of the flowchart even G3 in addition to size > 1.7 cm should undergo surgery. Similarly, I would recommend to add how to manage G1 and G2 gastric NENs (in the left part of the flowchart?), depending on the findings from the Authors’ analysis of the current literature.

According to the suggestion before, we edit the figure 2 adding the management based on the grading

-          Concerning Figure 2, I recommend to include in the flowchart what to do if histology after endoscopic treatment shows undetermined margins.

We specify in the yellow box with “positive margins”

Round 2

Reviewer 1 Report

I believe the manuscript has been improved and warrant publication after minor English editing.

Author Response

Dear Reviewer, thank for your review. 

We improve the english form of our manuscript. 

Regards

Doctor Alice Laffi 

Reviewer 2 Report

The title of the manuscript is: "Gastric neuroendocrine NEOPLASMS: a systematic review of management and outcomes of LOCALIZED type 3 g-NENs", but the manuscript is entirely dealing with only "NETs" (not NECs) and also with metastatic NETs (not only localized). I recommend to change the title accordingly.

Please, add in the “Methods” section what NIKE group means (‘NIKE’ = Neuroendocrine tumors Innovation Knowledge and Education?).

Regarding the flowchart in Figure 2, I strongly disagree on some major points, because the three prognostic factors emerged from your review (size, T stage, grade) are not correctly presented:

- In the flowchart, it appears that only tumor size is needed to decide for an endoscopic treatment. I recommend to include not only tumor size, but also grade and T(N) stage assessment (i.e. by endoscopic ultrasound) before "endoscopic resection" or "surgery".

- Tumor grade is put in the algorythm in a position that makes no sense. It seems that the choice on the systemic or surgical treatment depends only on stage (metastatic/localized/locally advanced), irrespective of tumor grade. I recommend to include tumor grade in a better position and to differentiate among grade 1, 2 and 3.

- Staging is also inserted in a non-sense position of the flowchart. EUS, CT scan/MRI, and PET/CT should be performed before treatment, but it appears as the staging process can follow the endoscopic resection, and that gNET with R0 endoscopic resection should not undergo a proper staging neither before nor after treatment. What about a small grade 3 gNET after a R0 endoscopic resection, should it undergo only endoscopic follow-up, or better a staging with morphological/functional imaging? I recommend to put staging in the first part of the flowchart.

- I recommend to change the last part of the discussion according to the previous points (rows 339 to 352).

Finally, the algorythm presented in Figure 2 in the present form is unclear and cannot correctly help in the decision-making process. If the Authors are not able to change it properly, I think it is better to completely delete it.

Author Response

We'd like to thank the reviewer for the observations. We hope that the changes that we have made to our manuscript aid to improve its quality. 

The title of the manuscript is: "Gastric neuroendocrine NEOPLASMS: a systematic review of management and outcomes of LOCALIZED type 3 g-NENs", but the manuscript is entirely dealing with only "NETs" (not NECs) and also with metastatic NETs (not only localized). I recommend to change the title accordingly.

We’d like to thank the reviewer for the observation. We edit the title accondringly.

Please, add in the “Methods” section what NIKE group means (‘NIKE’ = Neuroendocrine tumors Innovation Knowledge and Education?).

We specify the NIKE meaning in the methods.

Regarding the flowchart in Figure 2, I strongly disagree on some major points, because the three prognostic factors emerged from your review (size, T stage, grade) are not correctly presented:

- In the flowchart, it appears that only tumor size is needed to decide for an endoscopic treatment. I recommend to include not only tumor size, but also grade and T(N) stage assessment (i.e. by endoscopic ultrasound) before "endoscopic resection" or "surgery".

- Tumor grade is put in the algorythm in a position that makes no sense. It seems that the choice on the systemic or surgical treatment depends only on stage (metastatic/localized/locally advanced), irrespective of tumor grade. I recommend to include tumor grade in a better position and to differentiate among grade 1, 2 and 3.

- Staging is also inserted in a non-sense position of the flowchart. EUS, CT scan/MRI, and PET/CT should be performed before treatment, but it appears as the staging process can follow the endoscopic resection, and that gNET with R0 endoscopic resection should not undergo a proper staging neither before nor after treatment. What about a small grade 3 gNET after a R0 endoscopic resection, should it undergo only endoscopic follow-up, or better a staging with morphological/functional imaging? I recommend to put staging in the first part of the flowchart.

- I recommend to change the last part of the discussion according to the previous points (rows 339 to 352).

Finally, the algorythm presented in Figure 2 in the present form is unclear and cannot correctly help in the decision-making process. If the Authors are not able to change it properly, I think it is better to completely delete it.

We review the flowchart and we thank the reviewer for the suggestion. We edit the flowchart according the observation

Round 3

Reviewer 2 Report

The Authors addressed well the suggestions. I still have some major concerns regarding the final flowchart, that the Authors introduced as a summary of the literature findings that should guide the decision making process when dealing with gNETs.

-          In the last part of the discussion (rows 313-315), the Authors stated: “… based of size (T stage), grade and gastric wall infiltration”. Since T stage is both assessed by tumor size and gastric wall infiltration, I recommend to correct that sentence as follows: “… based of T stage (size and gastric wall infiltration) and grade”.

-          I suggest to start the flowchart (Figure 2) with the term “endoscopic biopsy” instead of “endoscopic detection”, to clearly show that histological features must be determined first, and please add “cm” to the tumor size. Then, based on tumor size and grade, the staging process seems to be the same irrespective of tumor size, so it is not needed to repeat the same staging modalities twice (EUS, CT/MRI, PET/CT).

-          According to the TNM stage of gastric NENs (ENETS TNM 2006, and UICC 2017 - 8^ Ed.), invasion of the submucosa (and muscularis mucosae) is considered T1, whereas invasion of the muscularis propria means T2. The Authors stated several times in the manuscript (abstract, results and discussion sections) that “muscularis mucosae” infiltration is a prognostic factor, so one could deduce that even T1 gNETs should be treated aggressively. Then, in the second part of the flowchart the use of the term “muscularis mucosae” is confusing, because T1-T2 gNETs are considered together (first box from the left), and also T3-T4 (and “muscularis mucosae”) are put together (second box from the left), so the subsequent treatment is not adequate. I recommend to check if the use of the term “muscularis mucosae” along the manuscript was intentional, or if a mistake occurred and the Authors meant “muscularis propria” instead. Therefore, I recommend to correct the flowchart accordingly to avoid any confusion. I suggest to first discriminate between “muscularis propria” infiltration or not (T1 vs. T2), since it decides among endoscopic resection (EMR/ESD) and surgery. Then, T2-T3 should be treated by surgery, whereas for T4 gNETs (with invasion of serosa or other organs), surgical and systemic treatment should be both considered.

I warmly recommend NOT to include the flowchart in the paper, if the Authors cannot manage to change and improve it properly.

Author Response

Point by point Reviewer 2 round 3

The Authors addressed well the suggestions. I still have some major concerns regarding the final flowchart, that the Authors introduced as a summary of the literature findings that should guide the decision making process when dealing with gNETs.

-          In the last part of the discussion (rows 313-315), the Authors stated: “… based of size (T stage), grade and gastric wall infiltration”. Since T stage is both assessed by tumor size and gastric wall infiltration, I recommend to correct that sentence as follows: “… based of T stage (size and gastric wall infiltration) and grade”.

We edit the manuscript accordingly

-          I suggest to start the flowchart (Figure 2) with the term “endoscopic biopsy” instead of “endoscopic detection”, to clearly show that histological features must be determined first,

We edit the flow-chart according to the suggestion.

and please add “cm” to the tumor size.

For the “cm”, it was a problem  of the size of the box accidentally occurred during the modifications. We edit the size of the box accordingly.

Then, based on tumor size and grade, the staging process seems to be the same irrespective of tumor size, so it is not needed to repeat the same staging modalities twice (EUS, CT/MRI, PET/CT).

We thank the reviewer for the observation. We edit the figure according the suggestion.

-          According to the TNM stage of gastric NENs (ENETS TNM 2006, and UICC 2017 - 8^ Ed.), invasion of the submucosa (and muscularis mucosae) is considered T1, whereas invasion of the muscularis propria means T2. The Authors stated several times in the manuscript (abstract, results and discussion sections) that “muscularis mucosae” infiltration is a prognostic factor, so one could deduce that even T1 gNETs should be treated aggressively. Then, in the second part of the flowchart the use of the term “muscularis mucosae” is confusing, because T1-T2 gNETs are considered together (first box from the left), and also T3-T4 (and “muscularis mucosae”) are put together (second box from the left), so the subsequent treatment is not adequate. I recommend to check if the use of the term “muscularis mucosae” along the manuscript was intentional, or if a mistake occurred and the Authors meant “muscularis propria” instead. Therefore, I recommend to correct the flowchart accordingly to avoid any confusion. I suggest to first discriminate between “muscularis propria” infiltration or not (T1 vs. T2), since it decides among endoscopic resection (EMR/ESD) and surgery. Then, T2-T3 should be treated by surgery, whereas for T4 gNETs (with invasion of serosa or other organs), surgical and systemic treatment should be both considered.

I warmly recommend NOT to include the flowchart in the paper, if the Authors cannot manage to change and improve it properly.

We thank to the reviewer for the really crucial observations. The term “muscularis mucosae” was a mistake derived from old manuscripts that we collected. We edit the flowchart and the main text according to the suggestion of the reviewer and we hope that this improves our work properly. Regarding the T4 stage, there is no a standard of care in terms of systemic treatment after a radical surgery in case of M0 and we preferred to not specify a further therapy.

Round 4

Reviewer 2 Report

The manuscript is now suitable for publication.